# Federated Online Clustering of Bandits

Xutong Liu[1]     Haoru Zhao[2]     Tong Yu[3]     Shuai Li[*2]     John C.S. Lui[1]

[1]The Chinese University of Hong Kong, Hong Kong SAR, China
[2]Shanghai Jiao Tong University, Shanghai, China
[3]Adobe Research, San Jose, CA, USA

## Abstract

Contextual multi-armed bandit (MAB) is an important sequential decision-making problem in recommendation systems. A line of works, called the clustering of bandits (CLUB), utilize the collaborative effect over users and dramatically improve the recommendation quality. Owing to the increasing application scale and public concerns about privacy, there is a growing demand to keep user data decentralized and push bandit learning to the local server side. Existing CLUB algorithms, however, are designed under the centralized setting where data are available at a central server. We focus on studying the federated online clustering of bandit (FCLUB) problem, which aims to minimize the total regret while satisfying privacy and communication considerations. We design a new phase-based scheme for cluster detection and a novel asynchronous communication protocol for cooperative bandit learning for this problem. To protect users' privacy, previous differential privacy (DP) definitions are not very suitable, and we propose a new DP notion that acts on the user cluster level. We provide rigorous proofs to show that our algorithm simultaneously achieves (clustered) DP, sublinear communication complexity and sublinear regret. Finally, experimental evaluations show our superior performance compared with benchmark algorithms.

## 1 INTRODUCTION

Stochastic multi-armed bandit (MAB) [Auer et al., 2002] is a well-known sequential decision-making problem, where a learner sequentially selects actions so as to maximize the cumulative rewards (or minimize the cumulative regret). One

fruitful application area of MAB is the online recommendation systems (RecSys) [Chu et al., 2011, Abbasi-Yadkori et al., 2011, Gentile et al., 2014, Li et al., 2019, Zhang et al., 2020, Li et al., 2021], where MAB algorithms provide a principled way to handle the challenge of exploration-exploitation trade-off [Lattimore and Szepesvári, 2020].

To advance the bandit algorithm for large-scale applications, contextual linear bandits add the simple yet effective linear structure assumptions on actions and reward functions [Chu et al., 2011, Li et al., 2010, Abbasi-Yadkori et al., 2011]. One limitation, however, is that such a model mainly works in a content-dependent manner, ignoring the often used tool of collaborative filtering. To address this issue, the clustering of bandits (CLUB) are proposed [Gentile et al., 2014, Li et al., 2016, Li and Zhang, 2018, Li et al., 2019]. The CLUB algorithms adaptively cluster similar users and utilize the collaborative information given by the cluster structure, which dramatically improves the recommendation quality.

While most existing bandit algorithms are designed under a centralized setting, in response to the increasing application scale and public concerns about privacy, there is a growing demand to keep user data decentralized and push the learning of bandit models to the client or the local server side . This paradigm is now known as federated learning [Kairouz et al., 2021]. Owing to its overall applicability, there has been a surge of interest in studying federated MAB [Dubey and Pentland, 2020, Zhu et al., 2021, Shi and Shen, 2021], which promises cooperative bandit learning with larger amounts of data (across multiple local servers) while keeping the data decentralized. This motivates us to study the CLUB problem to its federated counterpart, i.e., the federated clustering of bandits (FCLUB).

In FCLUB, each local server can conduct its own local clustering of bandit algorithms. To enable the collaborative effects of users across different servers, the local server could also collaborate with other local servers under the coordination of a global server, whose communication needs

---

[*]Correspondence to: Shuai Li <shuaili8@sjtu.edu.cn>

*Accepted for the 38[th] Conference on Uncertainty in Artificial Intelligence* (UAI 2022).

to satisfy specific privacy and communication requirements. The goal of this work is to design an federated online clustering of bandit framework, so as to minimize the $T$-round regret under the privacy protection requirements and communication cost considerations.

The key challenge of FCLUB is designing collaborative bandit learning procedures and cluster detection strategies to identify the overall cluster structures across different local servers, where each local server only holds part of the users with unknown interests. Such a problem is more challenging due to the following privacy and communication cost requirements, which are two first-order requirements for any federated applications [Kairouz et al., 2021].

**Privacy protection:** To reduce the privacy leakage of each user, we expect local servers to only share user clusters' data instead of individuals' raw data. In addition, we still need a mechanism to protect the uploaded (cluster) information against possible adversaries outside the local server, for which we adopt the solution concept of differential privacy (DP). However, the off-the-shelf DP notion is defined on individual users, hence unsuitable for FCLUB. It is challenging and unclear what is a suitable notion of privacy over the clustering of users and how to devise algorithms to guarantee the corresponding privacy requirements.

**Communication:** Communication is critical for collaborative learning, but may also be expensive or time-consuming. For FCLUB, it is desired to minimize the total regret while keeping the communication costs (in terms of communication rounds between the global server and local servers) as low as possible. Another requirement is to design an asynchronous communication protocol incorporating the randomly arriving users and possibly lagging servers, preventing commonly used synchronous protocols [Dubey and Pentland, 2020].

## 1.1 OUR CONTRIBUTIONS

To address the aforementioned challenges, this paper makes four contributions.

**1. Problem Formulation:** We propose the setting of online clustering of bandits to its federated counterpart, which considers the privacy protection and communication requirements. We also propose a novel cluster differential privacy (CDP) notion tailored for the FCLUB setting.

**2. Algorithm Design:** We propose a private and communication-efficient FCLUB-CDP algorithm. For privacy protection, a tree-based privatizer is designed to guarantee our proposed CDP. For communication efficiency, we follow the phase-based principle for cluster detection and propose the asynchronous communication protocol for delayed information sharing. In particular, each local server maintains upload/download buffers and occasionally up-

loads/downloads the buffered information to/from the global server only if it finds the latest information deviates too far from the last update.

**3. Theoretical Analysis:** We prove that FCLUB-CDP achieves the $O(dL\sqrt{mT\frac{\log(1/\delta)}{\varepsilon}}\log^{1.5}T)$ regret bounds, $O(dmL\log T)$ communication costs and $(\varepsilon, \delta, L, m)$-CDP privacy guarantee, respectively.

**4. Experiments:** We conduct extensive experiments over synthetic and real-world datasets to validate our theoretical analysis. Empirical results show the superior performance of our algorithm over existing algorithms.[1]

## 1.2 RELATED WORK

**Online Clustering of Bandits.** The online clustering bandits is first proposed by Gentile et al. [2014] and shows its effeteness by accelerating the learning process of contextual bandits. The key idea is to use a graph representing the user similarity and adaptively refine the user clusters for information sharing. This work has been extended by a series of works considering the collaborative effects on both users and items [Li et al., 2016], the context-aware settings [Gentile et al., 2017], the cascading bandit setting [Li and Zhang, 2018] and the users with different user frequency [Li et al., 2019]. However, none of these works consider the privacy constraints and communication cost requirements imposed by the FL paradigm like the current work, and therefore cannot give guarantees on these two critical criteria. Korda et al. [2016] considers the peer-to-peer but non-private clustering of bandits, our work studies the private bandit setting under the orchestration of a global server, which requires different algorithms and analysis.

**Federated and Distributed Bandits.** There has been growing interest in bandit learning with multiple players. One line of research investigates the competitive agents with collisions [Anandkumar et al., 2011, Rosenski et al., 2016, Bistritz and Leshem, 2018, Boursier and Perchet, 2019], in which the reward for an arm is zero if it is chosen by more than one agent. The goal of these works is to minimize regret without communication, which is different from ours. The cooperative distributed bandits are most related to our work, in which multiple agents collaborate to solve a bandit problem over certain communication networks, e.g., peer-to-peer networks [Korda et al., 2016] or client-server networks [Dubey and Pentland, 2020, Li and Wang, 2021]. Our work belongs to the client-server setting, but we differ from both Dubey and Pentland [2020] and Li and Wang [2021] since neither of them considers the clustering effects of users.

**Differential Privacy.** Our work leverages on *differential privacy*, a rigorous mathematical framework of privacy first

---

[1]Codes and datasets are available at GitHub.

proposed by Dwork et al. [2006]. We utilize several useful techniques from the standard differential privacy to maintain our cluster differential privacy condition. Most notably, we use a tree-based algorithm which is introduced in Chan et al. [2011] to realize differential privacy for the continual release of statistics. In the single-agent bandit setting, Shariff and Sheffet [2018] also utilizes this tree-based algorithm to achieve Joint DP, which is then extended by Dubey and Pentland [2020] to the federated setting. The closest work to ours is Dubey and Pentland [2020] and they study the simpler case where each local server only holds one user and all users are identical (with the same unknown preference vector), hence gives the different user-level DP definition with different privatizer and analysis.

*To the best of our knowledge, this paper is the first to generalize the CLUB to its federated setting, which simultaneously achieves privacy protection and communication requirements.*

## 2 PROBLEM SETTINGS

In this section, we formulate the setting of "Federated Clustering of Bandits" (FCLUB). We use $[n]$ to represent set $\{1, ..., n\}$. We use boldface lowercase letters and boldface capitalized letters for column vectors and matrices, respectively. For the norms, $\|\boldsymbol{x}\|$ denotes the $\ell_2$ norm of vector $\boldsymbol{x}$. For any symmetric positive semi-definite (PSD) matrix $\boldsymbol{M}$ (i.e., $\boldsymbol{x}^\top \boldsymbol{M} \boldsymbol{x} \geq 0, \forall \boldsymbol{x}$), $\|\boldsymbol{x}\|_{\boldsymbol{M}} = \sqrt{\boldsymbol{x}^\top \boldsymbol{M} \boldsymbol{x}}$ denotes the matrix norm of $\boldsymbol{x}$ regarding matrix $\boldsymbol{M}$.

At the global level, there are $n$ users, denoted by the set $\mathcal{U} = \{u_1, ..., u_n\}$. Each user $i \in \mathcal{U}$ has an *unknown* preference vector $\boldsymbol{\theta}_i \in \mathbb{R}^d$ and for simplicity we assume $\|\boldsymbol{\theta}_i\| \leq 1$. Since users may have the same/similar preference vector, we assume there exists $m$ (unknown) different preference vectors, i.e., $|\{\boldsymbol{\theta}_1, ..., \boldsymbol{\theta}_n\}| = m$. Users with the same preference vector form an underlying cluster and we denote these $m$ (unrevealed) clusters by $\mathcal{C} = \{C_1, ..., C_m\}$. Different from CLUB, users in FCLUB are distributed in $L$ local servers denoted by $\{1, ..., L\}$. At the local level, the local server $\ell$ contains $n^\ell$ users $\mathcal{U}^\ell = \{u_1^\ell, ..., u_{n^\ell}^\ell\}$ (with $\bigcup_{\ell \in [L]} \mathcal{U}^\ell = \mathcal{U}$) and similarly, these $n^\ell$ users form local cluster $\mathcal{C}^\ell = \{C_1^\ell, ..., C_{m_\ell}^\ell\}$ where $m_\ell \leq m$.

The learning agent interacts with the bandit game as the follows. At each time $t$, a user $i_t \in [n]$ randomly arrives with probability $1/n$. Then $K$ items are generated to form a item set $\boldsymbol{D}_t$, where the feature of each item $\boldsymbol{x} \in \boldsymbol{D}_t$ is drawn independently from a fixed but unknown distribution $\rho$ over $\{\boldsymbol{x} \in \mathbb{R}^d : \|\boldsymbol{x}\| \leq 1\}$. The learning agent identifies the local server $\ell_t$ that $i_t$ belongs to and the current user cluster $j_t$ (detected by our algorithm) which $i_t$ lies in. The local server then recommends an item $\boldsymbol{x} \in \boldsymbol{D}_t$ to the user based on the aggregated information from cluster $j_t$. After $i_t$ receives the recommendation, the

learning agent receives a random reward $y_t \in [0, 1]$. Let $\mathcal{H}_t = \{i_1, \boldsymbol{x}_1, y_1, ..., i_{t-1}, \boldsymbol{x}_{t-1}, y_{t-1}, i_t\}$ be the historical information before time $t$. We assume the expectation of reward $y_t$ is linear in the feature vector $\boldsymbol{x} \in \boldsymbol{D}_t$ and the unknown preference vector $\boldsymbol{\theta}_{i_t}$, i.e., $\mathbb{E}_t[y_t|\boldsymbol{x}] = \boldsymbol{\theta}_{i_t}^\top \boldsymbol{x}$, and $\{y_t - \boldsymbol{\theta}_{i_t}^\top \boldsymbol{x}\}_{t=1,2,...}$ have sub-Gaussian tails $\sigma_0^2$.

Now we give some assumptions on preference vectors and item feature vectors. Note that all the assumptions follow the previous works Gentile et al. [2014, 2017], Li and Zhang [2018], Li et al. [2019].

**Assumption 1** (Gap between preference vectors). *For any two different preference vectors $\boldsymbol{\theta}_{i_1} \neq \boldsymbol{\theta}_{i_2}$, there is a fixed but unknown gap $\gamma > 0$ so that $\|\boldsymbol{\theta}_{i_1} - \boldsymbol{\theta}_{i_2}\| \geq \gamma$.*[2]

**Assumption 2** (Item regularity). *For item distribution $\rho$, there exists a known $\lambda_x > 0$ so that $\mathbb{E}_{\boldsymbol{x} \sim \rho}[\boldsymbol{x} \boldsymbol{x}^\top]$ is full rank with minimal eigenvalue $\lambda_x$. Meanwhile, for all time $t$, for any fixed unit vector $\boldsymbol{\theta} \in \mathbb{R}^d$, $(\boldsymbol{\theta}^\top \boldsymbol{x})^2$ has sub-Gaussian tail with variance $\sigma^2 \leq \frac{\lambda_x}{8 \log(4K)}$.*

**Learning Efficiency.** The goal of the learning agent is to accumulate as much reward as possible. Let the optimal item for user $i_t$ at time $t$ be $\boldsymbol{x}_{i_t}^* = \arg\max_{\boldsymbol{x} \in \boldsymbol{D}_t} \boldsymbol{\theta}_{i_t}^\top \boldsymbol{x}$. The learning performance is measured by the regret, defined as

$$R(T) = \mathbb{E}[\sum_{t=1}^T r_t] = \mathbb{E}[\sum_{t=1}^T (\boldsymbol{\theta}_{i_t}^\top \boldsymbol{x}_{i_t}^* - \boldsymbol{\theta}_{i_t}^\top \boldsymbol{x}_t)], \quad (1)$$

where $r_t$ is the regret at time $t$ and the expectation is taken over the randomness of the algorithm and the environment regarding the users $i_1, ..., i_T$ and the item sets $D_1, ..., D_T$.

In addition to the regret, privacy protection and the communication cost are two important criterion in federated learning. In this work, we aim to ensure that the user data are protected under privacy constraints and the communication complexity is low.

**Privacy Requirements.** To protect the user data, we introduce two privacy requirements. First, we desire the local server only uploads the user clusters' sufficient statistics (or clustered data) for the learning procedure, instead of individual users' raw data. Second, to protect the clustered data against third-party adversaries outside the local server, we adopt the notion of DP, which encodes the intuition that any observable output changes very little (in probability) when any input datum changes. Since existing DP notions are defined on *user-level* data Shariff and Sheffet [2018], Dubey and Pentland [2020], we introduce a new differential privacy requirement to protect the *cluster-level* data.

The contextual MAB problem involves two sets of variables against the adversaries outside the local server: the decision sets $\boldsymbol{D}_t$ and the observed rewards $y_t$. Since the users

---

[2]As previous works, this assumption can be relaxed by assuming the existence of two thresholds, one for the between-cluster distance $\gamma$, the other for the within-cluster distance $\|\boldsymbol{\theta}_{i_1} - \boldsymbol{\theta}_{i_2}\| \leq \eta$.

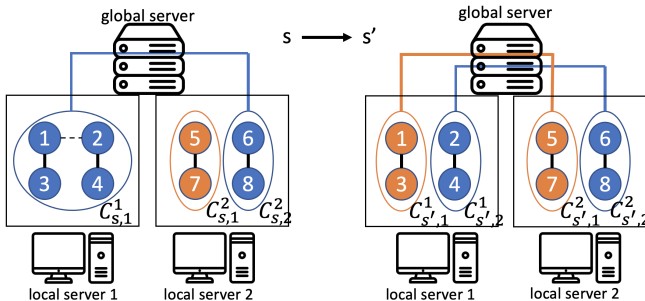

Figure 1: Illustration of how our algorithm detects clusters from phase $s$ to $s'$. Local server 1 delete edge $(1,2)$ and split $C_{s,1}^1$ into $C_{s',1}^1, C_{s',2}^1$. The global server merges local clusters as $\{(C_{s',1}^1, C_{s',1}^2), (C_{s',2}^1, C_{s',2}^2)\}$.

only receive and store observations regarding the chosen action $\boldsymbol{x}_t$ and the observed reward $y_t$, it suffices to protect $(\boldsymbol{x}_t, y_t)_{t\in[T]}$ to achieve DP requirements. Let $\tau_{\ell,j}$ be time slots when user in cluster $j$ appears at local server $\ell$, we denote two sequences $S_{\ell,j} = (\boldsymbol{x}_t, y_t)_{t\in\tau_{\ell,j}}$ and $S'_{\ell,j} = (\boldsymbol{x}'_t, y'_t)_{t\in\tau_{\ell,j}}$ as $t$-neighboring if $(\boldsymbol{x}_\tau, y_\tau) = (\boldsymbol{x}'_\tau, y'_\tau)$ for $\tau \neq t \in \tau_{\ell,j}$.

**Definition 1** (Cluster Differential Privacy). *In the FCLUB setting with $L$ servers and (at most) $m$ clusters, a federated contextual bandit algorithm $A = (A_{\ell,j})_{\ell\in[L],j\in[m]}$ is $(\varepsilon, \delta, L, m)$-CDP, if for any $(\ell,j),(\ell',j')$ s.t. $(\ell,j) \neq (\ell',j')$, any $t$ and the set of sequences $\bar{S}_{\ell',j'} = \cup_{i\in[L],k\in[m]} S_{i,k}$ and $\bar{S}'_{\ell',j'} = \cup_{i\in[L]\setminus\ell',k\in[m]\setminus j'} S_{i,k} \cup S'_{\ell',j'}$ s.t. $S_{\ell',j'}$ and $S'_{\ell',j'}$ are $t$ neighboring, and for any subset of actions $a_{\ell,j} \subset \prod_{\tau\in\tau_{\ell,j}} \mathcal{D}_\tau$ of actions, it holds that*

$$\Pr[A_{\ell,j}(\bar{S}_{\ell',j'}) \in a_{\ell,j}] \leq e^\varepsilon \Pr[A_{\ell,j}(\bar{S}'_{\ell',j'}) \in a_{\ell,j}] + \delta. \tag{2}$$

Note that our CDP notion formalizes the intuition that the action chosen by any local server $\ell$ (at the cluster level $j$) must be sufficiently indistinguishable (in probability) to any single $(x, y)$ pair from any other local cluster $(\ell', j')$. Such a notion does not require each cluster is private to its own observations, i.e., each cluster of users can be trusted with its own data, which is different from the local DP [Zheng et al., 2020] or Joint DP [Shariff and Sheffet, 2018] that assume even itself cannot be trusted.

**Communication Complexity.** To evaluate the communication complexity, we count one upload operation (or one download operation) between any local server and the global server as one communication round. Our communication complexity is the total number of communication rounds $C(T)$ over the time horizon $T$.

## 3 ALGORITHM

In this section, we introduce our phase-based federated clustering of bandit algorithms with CDP (FCLUB-CDP).

**Identify the Underlying Cluster Structure.** To correctly identify the cluster structure, we design a phase-based clustering detection algorithm in Algorithm 2. The high level idea is to first conduct local-level clustering of bandits on each local server and merge the local clusters on the global server.

At the local level, each server $\ell$ maintains a profile of $(\boldsymbol{V}_{t,i}, \boldsymbol{b}_{t,i}, T_{t,i})$ for its own local users $i \in \mathcal{U}^\ell$, where $V_{t,i}$ is Gramian matrix, $\boldsymbol{b}_{t,i}$ is the moment vector of regressand by the regressors, and $T_{t,i}$ is the number of times that $i$ has appeared up to time $t$. At the beginning of $s$ phase (or $t = 2^s + 1$), based on profiles $(\boldsymbol{V}_{t,i}, \boldsymbol{b}_{t,i}, T_{t,i})_{i\in\mathcal{U}^\ell}$, each server $\ell$ maintains an undirected graph structure $\mathcal{G}_s^\ell$, where nodes represent all local users $\mathcal{U}^\ell$ and a pair of users are connected by an edge if they are similar. We initialize the graph by a complete graph $\mathcal{G}_0^\ell$ and gradually delete edges at every beginning of each phase $s$. Specifically, in line 3, we delete edge between any user $i_1$ and $i_2$ if the distance between their estimated preference vector are larger than the following threshold.

$$\left\| \hat{\boldsymbol{\theta}}_{t,i_1} - \hat{\boldsymbol{\theta}}_{t,i_2} \right\| > \alpha_1(F(T_{t,i_1}) + F(T_{t,i_2})), \tag{3}$$

where $\hat{\boldsymbol{\theta}}_{t,i} = (\lambda\boldsymbol{I} + \boldsymbol{V}_{t,i})^{-1}\boldsymbol{b}_{t,i}$ and $F(x) = \sqrt{\frac{1+\ln(1+x)}{(1+x)}}$. After the deletion, users in connected components $j \in C(\mathcal{G}_s^\ell)$ are grouped into local cluster $j$. In line 5, server $\ell$ uploads the clustered information $I_{s,\ell} = (C_{s,j}^\ell, \tilde{\boldsymbol{V}}_{s,j}^\ell, \tilde{\boldsymbol{b}}_{s,j}^\ell, \tilde{T}_{s,j}^\ell)_{j\in C(\mathcal{G}_s^\ell)}$ to the global server, which contains clustered information for each cluster $j \in C(\mathcal{G}_s^\ell)$. Note that $I_{s,\ell}$ are added with random perturbation to protect the users' data, which will be introduced shortly after.

In line 7, when the global server receives the privatized clustered information from all servers, it performs a merge operation to merge clusters from different servers whose estimated clustered preference vectors are close into a global clusters according to the following inequality.

$$\left\| \hat{\boldsymbol{\theta}}_{s,j_1}^{\ell_1} - \hat{\boldsymbol{\theta}}_{s,j_2}^{\ell_2} \right\| < \alpha_2(F(\tilde{T}_{s,j_1}^{\ell_1}) + F(\tilde{T}_{s,j_2}^{\ell_2})), \tag{4}$$

where $\hat{\boldsymbol{\theta}}_{s,j}^\ell = (\tilde{\boldsymbol{V}}_{s,j}^\ell)^{-1}\tilde{\boldsymbol{b}}_{s,j}^\ell$ and $F(x) = \sqrt{\frac{1+\ln(1+x)}{(1+x)}}$. $P_s$ denotes the set of $m_s$ global clusters. Note that the local clusters in the same global cluster indexed by $k \in [m_s]$ will communicate and share protected clustered information with each other in an asynchronous manner. At the beginning of phase $s$, if the new global cluster structure $P_s$ is different from $P_{s-1}$ at phase $s-1$, we will renew the shared global information $(\boldsymbol{S}_{t,k}^g, \boldsymbol{u}_{t,k}^g, T_{t,k}^g)$ for $k \in [m_s]$. For local servers $(\ell, j)$ in the same global cluster $P_{s,k}$, the local synchronized information $(\boldsymbol{S}_{t,j}^\ell, \boldsymbol{u}_{t,j}^\ell, T_{t,j}^\ell)$, the upload buffers $(\Delta\boldsymbol{S}_{t,j}^\ell, \Delta\boldsymbol{u}_{t,j}^\ell, \Delta T_{t,j}^\ell)$ and download buffers $(\Delta\boldsymbol{S}_{t,j}^{-\ell}, \Delta\boldsymbol{u}_{t,j}^{-\ell}, \Delta T_{t,j}^{-\ell})$ are renewed. We also generate a new tree-based privatizer $\text{PVT}(\ell, j)$ for each cluster $j$ at server $\ell$, which will be introduced later on.

**Algorithm 1** Phase-based FCLUB with CDP

1: **Input:** Failure probability $\alpha$, deletion parameter $\alpha_1 > 0$, merge parameter $\alpha_2 > 0$, privacy parameters $\varepsilon, \delta$.

2: User initialization: For $i \in [n]$, $\boldsymbol{V}_{0,i} = \boldsymbol{0}_{d \times d}$, $\boldsymbol{b}_{0,i} = \boldsymbol{0}_{d \times 1}$, $\boldsymbol{T}_{0,i} = 0$.

3: Local server initialization: For $\ell \in [L]$, set graph $\mathcal{G}_0^\ell = (\mathcal{U}^\ell, \mathcal{E}_0^\ell)$, local information, upload buffers, download buffers: $(\boldsymbol{S}_{0,1}^\ell, \boldsymbol{u}_{0,1}^\ell, T_{0,1}^\ell) = (\Delta \boldsymbol{S}_{0,1}^\ell, \Delta \boldsymbol{u}_{0,1}^\ell, \Delta T_{0,1}^\ell) = (\Delta \boldsymbol{S}_{0,1}^{-\ell}, \Delta \boldsymbol{u}_{0,1}^{-\ell}, \Delta T_{0,1}^{-\ell}) = (\boldsymbol{0}_{d \times d}, \boldsymbol{0}_{d \times 1}, 0)$, perturbations $(\bar{\boldsymbol{H}}_{t,1}^\ell, \bar{\boldsymbol{h}}_{t,1}^\ell) = (\boldsymbol{H}_{t,1}^\ell, \boldsymbol{h}_{t,1}^\ell) = (\boldsymbol{0}_{d \times d}, \boldsymbol{0}_{d \times 1})$.

4: Global server initialization: create one global cluster $P_0 = \{\{(1, 1), ..., (L, 1)\}\}$ and set the global information for $V_{0,1}^g = \boldsymbol{0}_{d \times d}$, the $\boldsymbol{u}_{0,1}^g = \boldsymbol{0}_{d \times 1}$, $T_{0,1}^g = 0$.

5: **for** $s = 1, 2, ...,$ **do**

6:     Detect and adjust clusters (Algorithm 2).

7:     **for** $\tau = 1, ..., 2^s$ **do**

8:         Compute the total time step $t = 2^s - 2 + \tau$.

9:         Advance all parameter, e.g., $\boldsymbol{S}_{t,j}^\ell = \boldsymbol{S}_{t-1,j}^\ell$.

10:         User $i_t$ at local server $l_t$ arrives and $l_t$ gets the local cluster $j_t$ that $i_t$ belongs to based on $\mathcal{G}_s^\ell$.

11:         Compute local $\beta_{t,j_t}^{l_t}$ according to Lemma 1.

12:         Local server $l_t$ receives feasible context set $\boldsymbol{D}_t$ and recommends item $\boldsymbol{x}_t = \arg\max_{\boldsymbol{x} \in \boldsymbol{D}_t} \boldsymbol{x}^\top (\boldsymbol{S}_{t,j_t}^{l_t})^{-1} \boldsymbol{u}_{t,j_t}^{l_t} + \beta_{t,j_t}^{l_t} \|\boldsymbol{x}\|_{(\boldsymbol{S}_{t,j_t}^{l_t})^{-1}}$.

13:         User $i_t$ receives feedback $y_t$, update the user $i_t$'s information: $(\boldsymbol{T}_{t,i_t}, \boldsymbol{V}_{t,i_t}, \boldsymbol{b}_{t,i_t}) \mathrel{+}= (1, \boldsymbol{x}_t \boldsymbol{x}_t^\top, y_t \boldsymbol{x}_t)$ and others unchanged.

14:         Check upload event (Algorithm 3).

15:         Check download event (Algorithm 4).

16:     **end for**

17: **end for**

---

**Asynchronous Communication Protocol.** In this work, we design a novel asynchronous communication protocol to incorporate the randomly arriving users. To reduce the communication cost, our high-level idea is to use the delayed communication, where the feedback are temporarily stored in buffers and only if the stored information exceeds a threshold, the upload/download events are triggered. Such a threshold will ensure that the local information will not diverge too far from the global information, which in turn will not diverge too far from the scenario when information are fully synchronized. Also note that our communication is conducted in the asynchronous manner at the local cluster level. In other words, all local clusters indexed by $(\ell, j) \in P_{s,k}$ will establish connection with each other within the global cluster $k$. For each local cluster $(\ell, j)$, it stores a local copy of the sufficient statistics $(\boldsymbol{S}_{t,j}^\ell, \boldsymbol{u}_{t,j}^\ell, T_{t,j}^\ell)$ and a upload buffer $(\Delta \boldsymbol{S}_{t,j}^\ell, \Delta \boldsymbol{u}_{t,j}^\ell, \Delta T_{t,j}^\ell)$. For the global server, it prepares for each local cluster $(\ell, j)$ a download buffer $(\Delta \boldsymbol{S}_{t,j}^{-\ell}, \Delta \boldsymbol{u}_{t,j}^{-\ell}, \Delta T_{t,j}^{-\ell})$, which are used to send other local servers' information to the local cluster. It also maintains the global statistics $(\boldsymbol{S}_{t,k}^g, \boldsymbol{u}_{t,k}^g, T_{t,k}^g)$ to save the data

uploaded from local clusters in global cluster $k$.

Our proposed communication framework consists of two components: the upload protocol (Algorithm 3) and the download protocol (Algorithm 4). For the upload protocol, at each time step $t$, user $i_t$ visits the server $l_t$ and receives recommended item $\boldsymbol{x}_t$. After the user $i_t$ interacts with the environment and observes feedback $(\boldsymbol{x}_t, y_t)$, the local server updates the upload buffers in line 1 and checks the following condition to decide whether to upload the upload buffer:

$$\det(\boldsymbol{S}_{t,j_t}^{l_t} + \Delta \boldsymbol{S}_{t,j_t}^{l_t}) / \det(\boldsymbol{S}_{t,j_t}^{l_t}) \geq U, \tag{5}$$

where $\boldsymbol{H}_{t,j_t}^{l_t}$ and $\bar{\boldsymbol{H}}_{t,j_t}^{l_t}$ are tentative and current perturbation for privacy protection, respectively. If the condition is satisfied, the local server sends $(\Delta \boldsymbol{S}_{t,j_t}^{l_t}, \boldsymbol{u}_{t,j_t}^{l_t}, T_{t,j_t}^{l_t})$ to the global cluster $k_t$. The global server then merges the uploaded information into the global information in line 4 and also sends it to download buffers for other local clusters $(\ell, j) \neq (l_t, j_t) \in P_{s,k_t}$ in line 6. For local cluster $(l_t, j_t)$ itself, the local server updates the local statistics and initializes the upload buffer using the newly generated perturbation in lines 8 to 10. For the download protocol, at each time step $t$, the global server will check the deviation between global statistics and the local statistics via following condition:

$$\det(\boldsymbol{S}_{t,k_t}^g) / \det(\boldsymbol{S}_{t,j}^\ell) \geq D \tag{6}$$

independently for local clusters $(\ell, j)$ in global cluster $P_{s,k_t}$. If any cluster $(\ell, j)$ satisfies such condition, the global server sends the information from other clusters to $(\ell, j)$, which is used to update $(\ell, j)$'s local statistics. Finally, the global server cleans the download buffer.

**Tree-Based Privacy Protocol.** To ensure the uploaded information are privatized, we adopt the tree-based privatizer to generate random perturbations $\boldsymbol{H}_{t,j}^\ell$ and $\boldsymbol{h}_{t,j}^\ell$ whenever an upload event happens. Note that the privatier subroutine is at the local cluster level and a new privatizer is created if the cluster structure changes at the start of any phase $s$.

Let $\boldsymbol{x}_1, ..., \boldsymbol{x}_T$ be a (matrix-valued) sequence of length $T$, and $s_i = \sum_{t=1}^i \boldsymbol{x}_i$ be the partial sum of the first $i$ elements that will be realised privately. Generally speaking, the tree-based mechanism Dwork et al. [2006] maintains a binary tree $\mathcal{T}$ of depth $1 + \lceil \log T \rceil$, where the leaf nodes contain the elements $\boldsymbol{x}_i$ and the parent node maintains the sum of its children. For each node with value $n_i$, the tree-base mechanism protects privacy by adding noise $h_i$ to each node and release $n_i + h_i$ if queried. The key advantage is that such a tree only accesses $\nu = O(\log T)$ nodes to compute and release the partial sum $s_i$, which means the perturbation is at most $O(\nu)$ instead of $O(T)$.

Following this general idea, we implement the tree-based privatizer $(\ell, j)$ that satisfies the requirements of CDP. Recall that we only need to protect the information uploaded to the global server, it suffices to maintain a tree $\mathcal{T}_j^\ell$ of depth $\nu = O(1 + \lceil \log t_c \rceil)$ for the upload event, where $t_c$ is the

total number of uploads. To make the partial sums private, we insert a random noise matrix to each node in $\mathcal{T}_j^\ell$, similar to that of Shariff and Sheffet [2018] and Dubey and Pentland [2020]. Specifically, we sample a random matrix $\bar{N} \in \mathbb{R}^{(d+1)\times(d+1)}$ where each entry $\bar{N}_{p,q}$ is drawn from i.i.d. Gaussian distribution $\mathcal{N}(0, \sigma_{\text{noise}})$ and symmetrize it to get $N = (\bar{N}^\top + \bar{N})/\sqrt{2}$. It follows that in order to ensure the whole tree is $(\varepsilon, \delta)$-DP, each node should preserve $(\varepsilon/\sqrt{8\nu\log(2/\delta)}, \delta/2)$-DP. In other words, it suffices to set the variance $\sigma_{\text{noise}} = 64\nu\log(2/\delta)^2/\epsilon^2$ for each tree node. Note that at each upload round $t$, the total noise added to the partial sum is the summation of at most $\nu$ random matrices with size $(d+1) \times (d+1)$, where the top-left $(d \times d)$-submatrix forms $\boldsymbol{H}_{t,j}^\ell$ and the first $d$ elements from the right-most $(d+1) \times 1$ vector forms $\boldsymbol{h}_{t,j}^\ell$. By concentration of random matrices Tao [2011], we have with probability at least $(1 - \frac{\alpha}{mL})$, the operator norm of $\boldsymbol{H}_{t,j}^\ell$ is

$$\left\|\boldsymbol{H}_{t,j}^\ell\right\|_{op} \le \rho \triangleq 8\sqrt{2}\nu\log(4/\delta)(4\sqrt{d}+2\log(2mL/\alpha))/\varepsilon. \tag{7}$$

for any $\ell \in [L], j \in [m], t \in [T]$.

**Recommendation Procedure.** At each time step $t$, the recommended item $\boldsymbol{x}_t$ for user $i_t$ is selected as follows. When the current cluster is correct (which is guaranteed after $O(\log T)$ rounds and to be proved later), the estimated $\hat{\boldsymbol{\theta}}_t = (\boldsymbol{S}_{t,j_t}^{l_t})^{-1}\boldsymbol{u}_{t,j_t}^{l_t}$ is computed using the local information $\boldsymbol{S}_{t,j_t}^{l_t}$ and $\boldsymbol{u}_{t,j_t}^{l_t}$. Since by Lemma 1, $\left\|\boldsymbol{\theta}_{i_t} - \hat{\boldsymbol{\theta}}_t\right\|_2 \le \beta_{t,j_t}^{l_t}$, the confidence radius is $\beta_{t,j_t}^{l_t}\|\boldsymbol{x}\|_{(\boldsymbol{S}_{t,j_t}^{l_t})^{-1}}$, which characterizes the exploration bonus for item $\boldsymbol{x} \in \boldsymbol{D}_t$. Then the local server will recommend the item $\boldsymbol{x}_t \in \boldsymbol{D}_t$ that maximizes the $\boldsymbol{x}^\top \hat{theta}_t$ plus the above exploration bonus. Finally, the user will receive feedback $y_t$ and the system updates corresponding statistics for better decision in future rounds.

**Lemma 1.** *Under the setting of FCLUB and fix a local cluster $j$ located at the server $\ell$ which shares the information with $L' \le L$ clusters (including itself), let the true preference vector be $\boldsymbol{\theta}^*$ and the true cluster be $j^*$, let $\hat{\boldsymbol{\theta}}_{t,j}^\ell = (\boldsymbol{S}_{t,j}^\ell)^{-1}\boldsymbol{u}_{t,j}^\ell$. When all (global) clusters are correctly identified and partitioned, it holds with probability at least $1 - 2\alpha$,*

$$\left\|\boldsymbol{\theta}^* - \hat{\boldsymbol{\theta}}_{t,j}^\ell\right\|_{\boldsymbol{S}_{t,j}^\ell} \le \beta_{t,j}^\ell, \tag{8}$$

*where* $\beta_{t,j}^\ell \triangleq \beta_j^\ell(T_{t,j}^\ell, L, \alpha/(mL)) = \sigma_0\sqrt{2\log(\frac{mL}{\alpha}) + d\log(\frac{\rho_{\max}}{\rho_{\min}} + \frac{T_{t,j}^\ell}{dL'\rho_{\min}})} + \sqrt{L'\rho_{\max}} + \sqrt{L'\kappa}$.

## 4 RESULTS

Recall that perturbations $\bar{\boldsymbol{H}}_{t,j}^\ell, \bar{\boldsymbol{H}}_{t,j}^\ell$ are designed to satisfy the $(\varepsilon, \delta, L, m)$-CDP requirement. In particular, the privacy

---

**Algorithm 2** Phase-based Cluster Detection and Adjustment

1: $t = 2^s - 1$.
2: **for** $\ell \in [L]$ **do**
3:    Set $\mathcal{G}_s^\ell$ by deleting any edge $(i_1, i_2) \in \mathcal{G}_{s-1}^\ell$ if Equation (3) holds.
4:    For $\ell \in [L], j \in C(\mathcal{G}_s^\ell)$, generate new perturbation $\boldsymbol{H}_{t,j}^\ell, \boldsymbol{h}_{t,j}^\ell$ using PVT$(\ell, j)$ in Algorithm 5 and set historical $\bar{\boldsymbol{H}}_{t,j}^\ell = \boldsymbol{H}_{t,j}^\ell, \boldsymbol{h}_{t,j}^\ell = \bar{\boldsymbol{h}}_{t,j}^\ell$.
5:    For $\ell \in [L]$, upload the local clustered information $I_{s,\ell} = (C_{s,j}^\ell, \tilde{\boldsymbol{V}}_{s,j}^\ell, \tilde{\boldsymbol{b}}_{s,j}^\ell, \tilde{T}_{s,j}^\ell)_{j \in C(\mathcal{G}_s^\ell)}$ to the global server, where $(\tilde{\boldsymbol{V}}_{s,j}^\ell, \tilde{\boldsymbol{b}}_{s,j}^\ell, \tilde{T}_{s,j}^\ell) = (2\rho\boldsymbol{I} + \boldsymbol{H}_{t,j}^\ell, \boldsymbol{h}_{t,j}^\ell, 0) + \sum_{i \in C_{s,j}^\ell}(\boldsymbol{V}_{t,i}, \boldsymbol{b}_{t,i}, T_{t,i})$.
6: **end for**
7: The global server does global merge based on $I_s$ and get $m_s$ global clusters $P_s = \{P_{s,1}, ..., P_{s,m_s}\}$, where the two local clusters $C_{t,j_1}^{\ell_1}, C_{t,j_2}^{\ell_2}$ (with $\ell_1 \ne \ell_2$) are merged together in $P_{s,k}$ if Equation (4) holds.
8: **if** $s = 0$ or $P_s \ne P_{s-1}$ **then**
9:    //Renew the cluster information.
10:    **for** $k \in [m_s]$ **do**
11:       Set global gram matrix $(\boldsymbol{S}_{t,k}^g, \boldsymbol{u}_{t,k}^g, T_{t,k}^g) = \sum_{(\ell,j) \in P_{s,k}}(\tilde{\boldsymbol{V}}_{s,j}^\ell, \tilde{\boldsymbol{b}}_{s,j}^\ell, \tilde{T}_{s,j}^\ell)$.
12:       **for** $(\ell, j) \in P_{s,k}$ **do**
13:          Set $(\boldsymbol{S}_{t,j}^\ell, b_{t,j}^\ell, T_{t,j}^\ell) = (\boldsymbol{S}_{t,k}^g, \boldsymbol{b}_{t,k}^g, T_{t,k}^g)$.
14:          Create new perturbation $\boldsymbol{H}_{t,j}^\ell, \boldsymbol{h}_{t,j}^\ell$ using PVT$(\ell, j)$ in Algorithm 5.
15:          Set new $(\Delta\boldsymbol{S}_{t,j}^\ell, \Delta\boldsymbol{u}_{t,j}^\ell, \Delta T_{t,j}^\ell) = (3\rho\boldsymbol{I} + \boldsymbol{H}_{t,j}^\ell - \bar{\boldsymbol{H}}_{t,j}^\ell, \boldsymbol{h}_{t,j}^\ell - \bar{\boldsymbol{h}}_{t,j}^\ell, 0)$
16:          Set new $(\Delta\boldsymbol{S}_{t,j}^{-\ell}, \Delta\boldsymbol{u}_{t,j}^{-\ell}, \Delta T_{t,j}^{-\ell}) = (\boldsymbol{0}, \boldsymbol{0}, 0)$.
17:       **end for**
18:    **end for**
19: **end if**

---

budget $(\varepsilon, \delta)$ affects the regret and communication bounds via the following quantities $(\rho_{\max}, \rho_{\min}, \kappa)$, which can be treated as spectral bounds for $\bar{\boldsymbol{H}}_{t,j}^\ell, \bar{\boldsymbol{H}}_{t,j}^\ell$. Let $\tilde{\boldsymbol{H}}_{t,j}^\ell = 2\rho\boldsymbol{I} + 3\rho c_{j,t}^\ell\boldsymbol{I} + \bar{\boldsymbol{H}}_{t,j}^\ell$, where $c_{j,t}^\ell$ is the number of uploads for local server $\ell$ and cluster $j$.

**Definition 2** (Approximately-accurate $\rho_{\min}, \rho_{\max}$ and $\kappa$). *The bounds $0 < \rho_{t,\min} \le \rho_{t,\max}$ and $\kappa > 0$ are $(\alpha/(mL))$-accurate for $(\bar{\boldsymbol{H}}_{t,j}^\ell)$ for any $\ell \in [L], j \in [m]$ and $t \in [T]$:*

$$\left\|\tilde{\boldsymbol{H}}_{t,j}^\ell\right\|_{op} \le \rho_{\max}, \left\|(\tilde{\boldsymbol{H}}_{t,j}^\ell)^{-1}\right\|_{op} \le \frac{1}{\rho_{\min}}, \left\|\bar{\boldsymbol{h}}_{t,j}^\ell\right\|_{(\tilde{\boldsymbol{H}}_{t,j}^\ell)^{-1}} \le \kappa \tag{9}$$

with probability at least $(1 - \frac{\alpha}{mL})$.

As will be shown later, our communication protocol ensures $c_{j,t}^\ell \in [0, d\log T/\log(\min\{U, D\})]$, so $\rho_{\min} = \rho$, $\rho_{\max} = 3\rho + 3\rho d\log T/\log(\min\{U, D\})$, and $\kappa = \|\bar{\boldsymbol{h}}_{t,j}^\ell\|/\sqrt{\rho}$, where $\rho \triangleq 8\sqrt{2}\nu\log(4/\delta)(4\sqrt{d} + 2\log(2mL/\alpha))/\varepsilon$ is given by our privatizer.

In the following, we will give general regret and communi-

**Algorithm 3** Check Upload Event

1: Update upload buffer $(\Delta\boldsymbol{S}_{t,j_t}^{l_t}, \Delta\boldsymbol{u}_{t,j_t}^{l_t}, \Delta T_{t,j_t}^{l_t})$ += $(\boldsymbol{x}_t\boldsymbol{x}_t^\top, y_t\boldsymbol{x}_t, 1)$.

2: **if** $\det(\boldsymbol{S}_{t,j_t}^{l_t} + \Delta\boldsymbol{S}_{t,j_t}^{l_t})/\det(\boldsymbol{S}_{t,j_t}^{l_t}) \geq U$ **then**

3:     The global cluster finds $k_t$ so that $(l_t, j_t) \in P_{s,k_t}$.

4:     Update global information $(\boldsymbol{S}_{t,k_t}^g, \boldsymbol{u}_{t,k_t}^g, T_{t,k_t}^g)$ += $(\Delta\boldsymbol{S}_{t,j_t}^{l_t}, \Delta\boldsymbol{u}_{t,j_t}^{l_t}, \Delta T_{t,j_t}^{l_t})$.

5:     **for** $(\ell, j) \neq (l_t, j_t) \in P_{s,k_t}$ **do**

6:         Global server updates other servers' download buffer $(\Delta\boldsymbol{S}_{t,j}^{-\ell}, \Delta\boldsymbol{u}_{t,j}^{-\ell}, \Delta T_{t,j}^{-\ell})$ += $(\Delta\boldsymbol{S}_{t,j_t}^{l_t}, \Delta\boldsymbol{u}_{t,j_t}^{l_t}, \Delta T_{t,j_t}^{l_t})$.

7:     **end for**

8:     Local server $l_t$ updates the local statistics: $(\boldsymbol{S}_{t,j_t}^{l_t}, \boldsymbol{u}_{t,j_t}^{l_t}, T_{t,j_t}^{l_t})$ += $(\Delta\boldsymbol{S}_{t,j_t}^{l_t}, \Delta\boldsymbol{u}_{t,j_t}^{l_t}, \Delta T_{t,j_t}^{l_t})$.

9:     Local server $l_t$ sets $(\bar{\boldsymbol{H}}_{t,j_t}^{l_t}, \bar{\boldsymbol{h}}_{t,j_t}^{l_t}) = (\boldsymbol{H}_{t,j_t}^{l_t}, \boldsymbol{h}_{t,j}^{l_t})$ and creates new perturbation $\boldsymbol{H}_{t,j_t}^\ell, \boldsymbol{h}_{t,j_t}^\ell$ using the tree-based privatizer PVT$(l_t, j_t)$.

10:     Local server $l_t$ initializes the upload buffer using the new perturbation: $(\Delta\boldsymbol{S}_{t,j_t}^{l_t}, \Delta\boldsymbol{u}_{t,j_t}^{l_t}, \Delta T_{t,j_t}^{l_t}) = (3\rho\boldsymbol{I} + \boldsymbol{H}_{t,j}^{l_t} - \boldsymbol{H}_{t,j_t}^{l_t}, \boldsymbol{h}_{t,j_t}^{l_t} - \bar{\boldsymbol{h}}_{t,j_t}^{l_t}, 0)$.

11: **end if**

---

**Algorithm 4** Check Download Event

1: **for** $(l, j) \in P_{s,k_t}$ **do**

2:     **if** $\det(\boldsymbol{S}_{t,k_t}^g)/\det(\boldsymbol{S}_{t,j}^\ell) \geq D$ **then**

3:         Local server receives $(\boldsymbol{S}_{t,j}^\ell, \boldsymbol{u}_{t,j}^\ell, T_{t,j}^\ell)$ += $(\Delta\boldsymbol{S}_{t,j}^{-\ell}, \Delta\boldsymbol{u}_{t,j}^{-\ell}, \Delta T_{t,j}^{-\ell})$.

4:         Global server cleans the download buffer: $(\Delta\boldsymbol{S}_{t,j}^{-\ell}, \Delta\boldsymbol{u}_{t,j}^{-\ell}, \Delta T_{t,j}^{-\ell}) = (\boldsymbol{0}, \boldsymbol{0}, 0)$.

5:     **end if**

6: **end for**

---

cation bounds using $(\rho_{\max}, \rho_{\min}, \kappa)$ and replace them with their exact values.

## 4.1 REGRET BOUND

We give the following theorem as our main result for the regret bound.

**Theorem 1.** *Suppose the cluster structure over the users and items satisfy the assumptions in Section 2 with gap parameter $\gamma > 0$ and item regularity parameter $1 \geq \lambda_x > 0$. If the privatizer produces random perturbation that are $(1/(8mLT))$-accurate as in Definition 2, with probability at least $1 - 1/T$, the regret is upper bounded by*

$$R(T) \leq \tilde{O}\Big(n\big(\frac{\log T}{\lambda_x^2} + \frac{\sigma_0^2 d \log T}{\lambda_x \gamma^2} + \frac{\log(1/\delta) \log T}{\lambda_x \varepsilon \gamma^2}\big) + dL\sqrt{mT\frac{\log(1/\delta)}{\varepsilon}}\log^{1.5} T\Big) \quad (10)$$

We will give the proof sketch for the above theorem 1.

---

**Algorithm 5** Privatizer PVT$(\ell, j)$ for cluster $j$ at server $\ell$

1: **Input:** Privacy budget $\varepsilon, \delta$, number of uploads $t_c$.

2: Create a binary tree $\mathcal{T}$ of depth $\nu = \lceil \log(t_c + 1) \rceil + 1$.

3: For each node, we generate a perturbation matrix matrix $N \in \mathbb{R}^{(d+1)\times(d+1)}$, where $\boldsymbol{N} = (\bar{\boldsymbol{N}} + \bar{\boldsymbol{N}}^\top)/\sqrt{2}$ and $\bar{\boldsymbol{N}} \in \mathbb{R}^{(d+1)\times(d+1)}$ with $\bar{N}_{p,q} \sim \mathcal{N}(0, 64\nu\frac{\log(2/\delta)^2}{\varepsilon^2})$.

4: Calculate a queue of $\mathcal{Q} = (\boldsymbol{H}_i, \boldsymbol{h}_i)_{i=1,\dots,t_c+1}$ for partial sums $s_0, \dots, s_{t_c}$.

5: Sequentially pop one pair of $\mathcal{Q}$ if PVT$(\ell, j)$ is called.

---

*Proof.* Our proof mainly consists of two parts. The first part bounds the number of exploration rounds $2T_0$ after which the overall user clusters are correctly detected at the global server. The second part is to bound the regret for the asynchronous contextual linear bandits after the clusters are partitioned correctly.

Different from standard online clustering bandits, the key technical challenge is to take care of the additional random Gaussian noise produced by the privater, which perturbs the true observation that is needed for global cluster detection and the regret analysis for contextual linear bandits. Moreover, such perturbed observation are also lagged behind the instant observation, since FCLUB-CDP adopts the "delayed" asynchronous communication where upload and download are triggered occasionally. This makes standard contextual bandit analysis no longer works and requires new proof techniques to handle the gap between instant observation and the lagged (and perturbed) observation.

For the first cluster detection part, by the assumption of item regularity, we prove that after $t \geq O(n(\frac{\log T}{\lambda_x^2} + \frac{d\sigma_0^2 \log T}{\lambda_x \gamma^2}))$ rounds, the local estimates are accurate enough so that the local clusters are correctly identified, similar to that of Li and Zhang [2018]. Specifically, the 2-norm distance between local estimate $\hat{\theta}_{t,i}$ and the truth $\theta_i$ for any user $i$ is less than $\gamma/4$. Thus the local clusters are split correctly for all local servers. Now for the global cluster detection, the global server receives the aggregated observation from correctly partitioned local clusters, in which random Gaussian noises are added. Based on spectra property of Gaussian noise matrices (definition 2), the global server will spend additional $O(\frac{n \log(1/\delta) \log T}{\lambda_x \varepsilon \gamma^2})$ rounds so that the perturbed estimate $\hat{\theta}_{s,j}^l$ are accurate enough at the beginning of phase $s = \lceil \log_2 T_0 \rceil$, where $T_0 = O(n(\frac{\log T}{\lambda_x^2} + \frac{d\sigma_0^2 \log T}{\lambda_x \gamma^2} + \frac{\log(1/\delta) \log T}{\lambda_x \varepsilon \gamma^2}))$. Therefore, after $t > 2T_0$, the overall user clusters are partitioned correctly.

For the regret after $2T_0$, we use the delayed update technique from [Abbasi-Yadkori et al., 2011, Section 5.1], which only recomputes the confidence radius only $O(\log T)$ times and hence saves computation. The same strategy can also be applied for the delayed communication. The key analysis relies on using the upload and download condition in eq. (5) and eq. (6), so that the actually-used cluster confidence

radius is at most $\Gamma$ times larger than that if all local servers upload their perturbed observations in a fully synchronized manner, where $\Gamma = \sqrt{D(1 + (L-1)(U-1)) + U - 1}$. This will give a $\Gamma R_j(T_j)$ regret for the second part, where $R_j(T_j)$ is the private-version regret for the cluster $j$ if all observation are synchronized at each round. The full proof is put in Appendix B. ∎

## 4.2 COMMUNICATION COST

We give the following theorem to bound the total communication cost.

**Theorem 2.** *Under the CDP setting, the total communication cost satisfies:*

$$C(T) \leq O(\frac{dmL \log T}{\log(\min\{U, D\}}) \tag{11}$$

*Proof.* The total communication cost also has two parts: the upload at the beginning of each phase for global cluster detection and the asynchronous communication within each phase for information sharing. For the first part, the algorithm has at most $\log T$ phases and at each phase, there are total $mL$ local clusters uploading the clustered information, hence the total communication cost is $O(mL \log T)$. For the second part, recall that we adopt the delayed asynchronous communication protocol and the total number of uploads and downloads can be bounded by $O(dmL \log T)$. See Appendix C for the detailed proofs. ∎

## 4.3 PRIVACY GUARANTEE

**Theorem 3.** *Algorithm 1 preserves $(\varepsilon, \delta, L, m)$-CDP as defined in Definition 1.*

*Proof.* The CDP condition is satisfied by assigning the right amount of Gaussian noise in each tree node of our tree-based privacy protocol in Section 3. See Appendix D for details. ∎

## 4.4 DISCUSSION AND COMPARISON

**Discussion on the Regret Bounds.** For the regret bound, our result has two terms: the regret before the clusters are correctly partitioned $n(\frac{\log T}{\lambda_x^2} + \frac{\sigma_0^2 d \log T}{\lambda_x \gamma^2} + \frac{\log(1/\delta) \log T}{\lambda_x \varepsilon \gamma^2})$ and the regret after the clusters are correctly partition $O(dL\sqrt{mT\frac{\log(1/\delta)}{\varepsilon}} \log^{1.5} T)$. We will compare our results with several degenerate cases, given that we are the first work to study the federated clustering of bandits setting. For these cases, the additional CDP causes at most $O(\sqrt{\frac{\log(1/\delta)}{\varepsilon}})$ factor and asynchronous communication protocol causes at most $O(\sqrt{dL \log T})$ factor in general.

First, when $m = 1, L = 1$, our setting degenerates to the linear bandits with DP where all users share the same underlying parameter. Compared to Shariff and Sheffet [2018] which gives a $O(\sqrt{d\frac{\log(1/\delta)}{\varepsilon}}\sqrt{T}\log^{1.5} T)$ regret with 0 communication, our bound has a $O(\sqrt{d})$ additional factor (or more precisely $O(\sqrt{d \log \log T})$ factor) for the second term, which stems from the larger perturbation in order to protect total $O(d \log T)$ communication rounds.

Second, when $L = 1$, our setting reduces to the online clustering bandits with DP, Li and Zhang [2018] gives a $O(n(\frac{\log T}{\lambda_x^2} + \frac{d\sigma_0^2 \log T}{\lambda_x \gamma^2}) + d\sqrt{mT} \log T)$ for the non-DP version. Since CDP mechanism requires random perturbation, the clustering process suffers an additional $\frac{n \log T \log(1/\delta)}{\lambda_x \varepsilon \gamma^2}$ for the first term and the second regret term now has a new $\sqrt{\frac{\log(1/\delta) \log T}{\varepsilon}}$ leading factor due to the CDP requirements.

Third, when $m = 1$ and if we consider the special case when each local server only has one user and all users come in a round-robin manner, our setting reduces to the distributed linear bandits with DP. Dubey and Pentland [2020] provides a synchronized algorithm that achieves $O(\sqrt{dLT\frac{\log(1/\delta)}{\varepsilon}}\log^{1.5} T)$, our second term has an additional $\sqrt{dL}$ factor because of different communication protocol, which enables asynchronous communication at the cost of the larger $O(dL \log T)$ (compared with $O(L \log T)$) communication rounds and an additional $\Gamma = O(\sqrt{L})$ factor in the confidence radius.

Finally, there is a lower bound $\Omega(\sqrt{dmT})$, if we consider the case where the clustering structure is known, the communication and privacy budgets are unlimited and each cluster contains equal number of users. In this case, it is equivalent to learn $m$ independent linear bandits, each with expected rounds $T/m$ and according to Dani et al. [2008], the lower bound is $\Omega(\sum_{i \in [m]} \sqrt{dT/m}) = \Omega(\sqrt{dmT})$. In other cases, the regret lower bound will be greater and the lower bound $\Omega(\sqrt{dmT})$ still holds. Our regret bound matches the lower bound up to a factor of $O(L\sqrt{d\frac{\log(1/\delta)}{\varepsilon}}\log^{1.5} T)$.

**Discussion on the Communication Cost.** Our communication cost also has two terms: the first $O(mL \log T)$ term for identifying clusters at the beginning of each phase and the leading $O(\frac{dmL \log T}{\log(\min\{U, D\}})$ term for our asynchronous communication protocol. Compared with Dubey and Pentland [2020] when $m = 1$ and users come at the round-robin manner, our communication has an additional $O(d)$ factor. Due to the specialty of the user arrival, the same paper can achieve communication cost independent of $T$ at the cost of $O(\log(LT))$ additional factor in the regret. Though our total communication cost can not be reduced below $O(mL \log T)$ due to the first term, it will be interesting to consider whether the similar trade-off works for our asynchronous protocol in the future work.

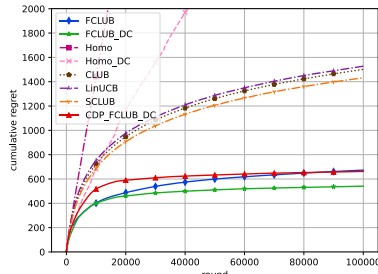
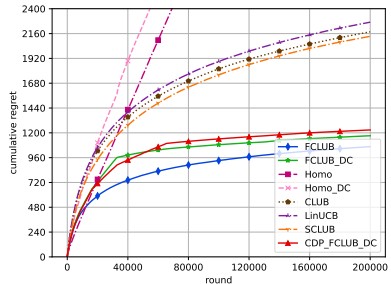

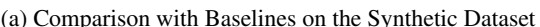

(a) Comparison with Baselines on the Synthetic Dataset    (b) Comparison with Baselines on the MovieLens Dataset

Figure 2: Comparative Experiments on the Cumulative Regret with Baseline Algorithms

## 5 EXPERIMENTS

To validate our theoretical findings, we conduct experiments on a synthetic dataset and a real-world MovieLens dataset. Algorithm 1 is denoted as CDP-FCLUB-DC and we also present its non-private version FCLUB-DC and its non-private, synchronized version FCLUB (by instantly uploading the observations). The baselines include CLUB which uses a separate CLUB algorithm Gentile et al. [2014] for each local server; SCLUB which uses a separate SCLUB algorithm Li et al. [2019] for each server; and LinUCB which uses a separate LinUCB algorithm Abbasi-Yadkori et al. [2011] for each user. We also consider the synchronized and asynchronous version of Algorithm 1 (denoted as Homo and Homo-DC, respectively) by treating users are identical with the same preference vector. Note that all results are averaged over ten random seeds, and we provide mean results with one unit of standard derivation for each curve. Due to the space limit, we provide the detailed experiment settings (including data generation and processing) in Appendix E.1, the parameter study Appendix E.2, the communication cost Appendix E.3 and running time results in Appendix E.4, respectively.

**Synthetic Dataset.** We first conduct experiments on a synthetic dataset. In Figure 2a, we compare our algorithm CDP-FCLUB-DC with the baselines listed above. The vertical axis indicates the cumulative regret and the horizontal axis indicates the round $t$. In general, our algorithm CDP-FCLUB-DC's performance has a clear advantage over baseline SCLUB, CLUB, LinUCB, Homo and Homo-DC. Since Homo-DC and Homo assumes users are in the same cluster, they mistakenly merge different clusters and suffer linear regrets, indicating the correctness of cluster detection is essential to have small regrets. Compared with SCLUB and CLUB that only perform local clustering operations, we can verify the correctness of our algorithm's clustering operations at the global level, which successfully leverages the collaborative effects across different local servers. As expected, CDP-FCLUB-DC performs a little worse than FCLUB and FCLUB-DC due to the delayed communication and cluster differential privacy requirements.

**MovieLens Dataset.** In this section, we also compare our algorithm CDP-FCLUB-DC with the baselines listed above on movie recommendations with the MovieLens dataset. The performances are shown in Figure 2b. Our algorithm CDP-FCLUB-DC's performance has an advantage over baseline SCLUB, CLUB, LinUCB, Homo and Homo-DC in general. Figure 2b shows CDP-FCLUB-DC performs worse than FCLUB and FCLUB-DC due to the delay communication and cluster differential privacy (CDP) as we have explained in synthetic dataset part. Different from the synthetic dataset, in the early stage, our algorithm needs more time to identify the underlying cluster structure. But after all user clusters are correctly detected at the global server, our algorithm performs better than Homo/Homo-DC that assume users are homogeneous, CLUB/SCLUB on each local server and LinUCB on each user.

## 6 CONCLUSION AND FUTURE WORK

In this paper, we formulate the federated online clustering of bandits problem, which generalizes the clustering of bandits problem to its federated counterpart. To tackle this new problem, we propose a FCLUB-CDP algorithm, which simultaneously achieves sublinear regret, sublinear communication complexity and satisfies our newly-defined clustered differential privacy requirements. Compared with benchmark algorithms, we show that FCLUB-CDP achieves superior performance regarding regret and communication cost. There are many compelling directions for future study. For example, it would be interesting to study our problem where local differential privacy is considered. One could also study a more efficient protocol to further reduce the communication cost.

### Acknowledgement

The corresponding author Shuai Li is supported by National Natural Science Foundation of China (62006151). This work is sponsored by Shanghai Sailing Program. The work of John C.S. Lui was supported in part by the RGC SRFS2122-4S02.

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
