# OpenReview forum: "Federated Online Clustering of Bandits"
_auai.org/UAI/2022/Conference — UAI 2022 Poster_

### Official Review · Reviewer_rPAK · 2022-03-31

**Q2(1) Originality/Novelty:** 3
**Q2(2) Significance/Impact:** 2
**Q2(3) Correctness/Technical Quality:** 3
**Q2(6) Clarity Of Writing:** 3
**Q6 Overall Score:** 6
**Q8 Confidence In Your Score:** 3

**Q1 Summary And Contributions:**

The paper studies the online clustering of linear bandits, with privacy and communication considerations, by leveraging the federated learning idea.
Multiple techniques from related areas are carefully combined, such as graph-based clustering, differential privacy, and an asynchronous communication protocol.
The desired performance is further supported by theoretical results and numerical experiments.


**Q10 Ethical Concerns (Optional):**

I do not have any major concerns.


**Q2 Assessment Of The Paper:**

More detailed information regarding each of these aspects is given below:

**Q2(4) Quality Of Experiments (Optional):**

3: Good: The experimental evaluation is adequate, and the results convincingly support the main claims.

**Q2(5) Reproducibility:**

2: Fair: Key resources (e.g., proofs, code, data) are unavailable but key details (e.g., proof sketches, experimental setup) are sufficiently well-described for an expert to confidently reproduce the main results.

**Q3 Main Strengths:**

1. The studied problem is meaningful and of potential practical impact.
2. The technical quality is good, with solid derivations and theoretical results.
3. The paper is clearly written (though a bit dense in notations).


**Q4 Main Weakness:**

1. No code is released, which might affect the reproducibility


**Q5 Detailed Comments To The Authors:**



**Q7 Justification For Your Score:**

Overall, the paper is a nice contribution to the bandit literature. A technically sound method is derived to address a meaningful problem, with rigorous theoretical guarantee and decent experimental supports. As a combination of several existing lines of research and (although non-trivial) of ideas developed therein, I think it is fair to say it has a `moderate` impact and hence I give 6.


**Q9 Complying With Reviewing Instructions:**

1: Yes.

---

### Official Review · Reviewer_f5c4 · 2022-04-12

**Q2(1) Originality/Novelty:** 3
**Q2(2) Significance/Impact:** 3
**Q2(3) Correctness/Technical Quality:** 3
**Q2(6) Clarity Of Writing:** 3
**Q6 Overall Score:** 5
**Q8 Confidence In Your Score:** 2

**Q1 Summary And Contributions:**

This paper focused on the clustering of bandits problem (CLUB) with the federated learning structure, which highly decreases the communication cost between the servers. Within this setting, the author proposed the Phase-based FCLUB with a CDP algorithm with $O(dL\sqrt{mT})$ regret guarantee.

**Q2 Assessment Of The Paper:**

More detailed information regarding each of these aspects is given below:

**Q2(4) Quality Of Experiments (Optional):**

3: Good: The experimental evaluation is adequate, and the results convincingly support the main claims.

**Q2(5) Reproducibility:**

3: Good: Key resources (e.g., proofs, code, data) are available and key details (e.g., proofs, experimental setup) are sufficiently well-described for competent researchers to confidently reproduce the main results.

**Q3 Main Strengths:**

This work utilizes the federated learning structure to decrease communication costs and provide a differential privacy guarantee for local information.

**Q4 Main Weakness:**

The assumption in section 2 seems too restrictive. It is not reasonable to assume that users with the same preference vector form an underlying cluster, which is somehow impractical from my point of view. A much more reasonable assumption is that the gap between preference vectors within a cluster is small than a threshold.

**Q5 Detailed Comments To The Authors:**

See Q4.

**Q7 Justification For Your Score:**

Applying a federated learning structure in the clustering of bandits problem is novel and important. However, the assumption that users with the same preference vector form an underlying cluster is too restrictive to be used in real-world applications.

**Q9 Complying With Reviewing Instructions:**

1: Yes.

---

### Official Review · Reviewer_6xCs · 2022-04-14

**Q2(1) Originality/Novelty:** 3
**Q2(2) Significance/Impact:** 3
**Q2(3) Correctness/Technical Quality:** 3
**Q2(6) Clarity Of Writing:** 3
**Q6 Overall Score:** 7
**Q8 Confidence In Your Score:** 3

**Q1 Summary And Contributions:**

The paper presents a study of the federated clustering of bandits with privacy and communication consideration in addition to the task of regret minimization. A new privacy notion tailored to this setting is also introduced.  An algorithm is proposed for the considered problem and performance guarantees on regret, communication costs and privacy are proven. Experimental results are given to support the theoretical results.

**Q2 Assessment Of The Paper:**

More detailed information regarding each of these aspects is given below:

**Q2(4) Quality Of Experiments (Optional):**

3: Good: The experimental evaluation is adequate, and the results convincingly support the main claims.

**Q2(5) Reproducibility:**

3: Good: Key resources (e.g., proofs, code, data) are available and key details (e.g., proofs, experimental setup) are sufficiently well-described for competent researchers to confidently reproduce the main results.

**Q3 Main Strengths:**

1. The considered problem setting is well-motivated and the introduced ideas seem novel.

2. The problem is treated extensively and the presented methods appear to be well-suited for the respective problems.

3. The paper is well-written. The mathematical details are easy to follow.



**Q4 Main Weakness:**

Lack of a lower bound.

**Q5 Detailed Comments To The Authors:**

This submission could be improved with the inclusion of a lower bound so it will be clear if the achieved upper bounds are optimal.


**Q7 Justification For Your Score:**

This paper introduces a new problem setting which is well motivated from practical applications. The validity of the provided solutions is validated with theoretical results and experimental results on real data. All things considered, I think this paper will be a good addition to the literature.

**Q9 Complying With Reviewing Instructions:**

1: Yes.

---

### Decision · Program_Chairs · 2022-05-15

**Decision:**

Accept (Poster)

**Comment:**

Meta Review: This paper studies the federated clustering of bandit (FCLUB) problem, whose task is to minimize the total regret meanwhile satisfying privacy and communication considerations.
The authors propose a scheme for cluster detection and asynchronous communication protocol for cooperative bandit learning and a modified differential privacy notion on user _cluster_ level.

They provided the theoretical analysis and showed that their algorithm achieved clustered differential privacy, sublinear communication complexity, and sublinear regret.

The reviewers found that the paper is solid. They gave questions on the lower bound of their regret, the validity of the assumption, and the code availability. The authors' feedback solved their concerns.

The average rating was 6. Two of the three reviewers gave the ratings >=6, and the other 5.